# Influence Factors of Small and Medium-Sized Enterprises and Micro-Enterprises in the Cross-Border E-Commerce Platforms

**Wei-Hung Chen** [1] , **Yao-Chin Lin** [2] **, Anima Bag** [2,*] **and Chun-Liang Chen** [3]

1   Department of Information Management, Chihlee University of Technology, New Taipei City 220305, Taiwan
2   Department of Information Management, Yuan Ze University, Taoyuan City 32003, Taiwan
3   Graduate School of Creative Industry Design, National Taiwan University of Arts,
    New Taipei City 22058, Taiwan
*   Correspondence: anima86.bag@gmail.com

**Abstract:** Small and medium-sized businesses (SMEs) are frequently exposed to a variety of difficulties during global epidemic crises like coronavirus (COVID-19), which may even threaten their lives. The purpose of this study explores the influencing factors of Taiwan's companies between small and medium-sized enterprises and micro-enterprises on the choice of the cross-border e-commerce platform. The findings are defined as taking into account small and medium-sized businesses and microenterprises when choosing cross-border e-commerce through a literature review and an examination of secondary data among the 10 participating businesses through interviews in various regions and business sectors in Taiwan. In this case we used study-based research, which included five small, medium-sized, and micro-enterprises, as well as five cross-border e-commerce projects and the company's management senior officers. According to the study's emphasis on the economic, social, technological, and legal aspects of various firms, these factors lead to a variety of decisions regarding the best cross-border e-commerce platform. The case study approach was utilized in this investigation to confirm the consideration of micro-and small-sized businesses that took part in cross-border e-commerce project counseling. This study summarizes five types of enterprises with different capabilities: product enhancement, marketing enhancement, cross-border potential, knowledge-based enhancement, and cross-border start-up. According to the results, it was found that different enterprise capabilities will affect the choice of cross-border e-commerce platforms. These five capabilities also have different types of consideration factors; among them, SMEs pay attention to marketing, pricing, market analysis, culture, customer service, payment, logistics, certification, taxation, etc. In addition to theoretical implications, this research also gives small and medium enterprises and micro-enterprises practice when choosing cross-border e-commerce platform, as well as suggestions for future research.

**Keywords:** small and medium enterprises; micro-enterprises; cross-border e-commerce platform; COVID-19; e-commerce

## 1. Introduction

Cross-border e-commerce (CBE), the newest and fastest-growing idea to contribute to a region in the global economy to drive based on e-commerce, is one of the greatest platforms in this COVID-19 predicament [1–3].

When choosing cross-border e-commerce, this refers to online deals. Transactions and payment are conducted in different countries and areas to deliver goods to complete the transaction process, along with logistics value chains [2]. Despite the contribution of CBE towards getting attention to maintain economic growth through change and increase market share to improve their products, for this pandemic, government policies support it [3]. CBE is an important platform that has the potential to gain trade growth. However, CBE has limited resources and capabilities to help small and medium-sized enterprises (SMEs) for

the international market [4]. Governments can develop policies to influence cross-border e-commerce to improve the further growth of small and medium-sized enterprises and micro-enterprises [5]. SMEs and micro-enterprises have a significant role in economy growth [6].

SMEs and micro-enterprises contribute to the standard work force in different developing countries and different industries [2,3]. Moreover, SMEs and micro-enterprises have an opportunity to explore their market process, products, and knowledge. For the COVID-19 situation, all of the developing countries' government policies support the growth of micro-enterprises, and promoting small and medium-sized enterprises has had a significant impact on the CBE market [7]. The feature of e-commerce can use cross-border business activities [8]. However, SMEs and micro-enterprises cannot build their platforms due to a lack of resources. Therefore, the strategies of selecting platforms allow for SMEs and micro-enterprises to apply cross-border e-commerce for sales.

Moreover, the government has a wide range of schemes to challenges and emerging opportunities for growth. In addition, the emergence of digital platforms is helping SMEs to reach international markets [9]. After that, SMEs may want to decide to start the CBE. The SME features will affect companies' decision-making to choose platforms for their cross-border e-commerce. The cross-border e-commerce platforms are established on internet technology by the network and trading factors [10]. The advancement of internet technology and e-commerce has gained quicker growth over the past few years. According to the Ministry of Economic Affairs (MOEA) report, the e-commerce market in Taiwan has jumped to 16.2% to reach $8.6 billion in 2020 [11].

In recent years, cross-border e-commerce platforms have developed more and more consumers choosing to collect foreign high-quality products [12], and services through the internet and direct overseas shopping, rather than the new foreign trade agent, are gradually being accepted by more and more people. Consumer attitudes change and promote the development of cross-border e-commerce [13]. The enterprises need to consider economic, social, technology, and legal factors issues when choosing a cross-border e-commerce platform. Several researchers differentiate barriers for cross-border e-commerce by analyzing and examining. Bingi et al. [14] pointed out several factors global e-commerce faces. The listed factors are considered challenges to e-commerce and are expected to raise the adoption of cross-border e-commerce.

The cost justification for the increasing expense of the global e-commerce capability to address anywhere in the world is made of economic elements [13,14]. The impact of social or cultural factors that comprise the trade parties are from different countries, and they speak different languages. In addition, the language barrier makes it difficult for the trade parties of cross-border e-commerce to easily communicate and understand each other [15]. The cultural factors to be faced are diversity, trust, and absence of touch and feel. Technology factors comprise internet access, infrastructure, and skill-shortage for important areas of challenges. Legal factors comprise privacy and security. However, there is still a lack of regulations in the field of cross-border e-commerce in Taiwan concerning the laws and regulations of foreign countries, which has limited development of cross-border e-commerce. A company intending to enter a country with cross-border e-commerce applications should consider the negative effects and barriers and take action to decrease them [16].

Despite this study focusing on SMEs, micro-enterprises, and cross-border e-commerce organizations to analyze barriers, product, marketing, and knowledge of corporate capabilities affect the factors of the cross-border e-commerce platform. In addition, this study aims to examine the Taiwanese SMEs and micro-enterprises focusing on cross-border e-commerce factors: economic, social, technology, and legal factors, respectively. Other factors are customer service, payment, logistics, certification, taxation, etc., which find out the consideration of SMEs or micro-enterprises on the choice of the CBE platform.

There is a lot of information about how small and medium-sized enterprises operate on cross-border e-commerce platforms in the existing domestic and foreign-related literature,

but there are fewer data about micro-enterprises. After all, there are still some differences between micro-enterprises and small and medium-sized enterprises. There will also be differences in the considerations for choosing a cross-border e-commerce platform, so this paper wants to know whether micro-enterprises and small and medium-sized enterprises face different issues when choosing a cross-border e-commerce platform, and hope to find a suitable one through this research. Micro-enterprises and SMEs each choose the model of cross-border e-commerce platform.

The purpose of this study explores the influencing factors for the choice of CBE platforms by SMEs and micro-enterprises. The research questions are as follows: What are the considerations factors of SMEs and micro-enterprises when choosing a CBE platform? How do enterprise capability characteristics affect the choice of CBE platforms in terms of product, marketing, and knowledge? What are the different factors between SMEs and micro-enterprises when choosing CBE? What are the influencing factors to consider?

## 2. Literature Review and Research Framework

### 2.1. SMEs and Micro-Enterprises

According to the international organizations of Taiwan's statistical data, generally each country has its own definition of 'SMEs and micro-enterprises', and there is no common worldwide definition due to each country's economic progress. The official definition varies widely from country to country, and can be specific to only one country, as is the case in Taiwan. The Ministry of Economic Affairs indicated that small and medium enterprises of Taiwan [17] are defined as manufacturing [18] and construction, are based on capital investments of up to NT$80 million, and have less than 200 regularly employed people. For instance, the turnover of the previous year report defined SMEs and micro-enterprises as industries that were below NT$100 million, where the number of regularly employed people was less than 100. The World Bank also defined SMEs and micro-enterprises companies that employ fewer than 100 employees.

The Organization of Economic Co-operation and Development (OECD) defined micro-enterprises as the smallest type of organization in terms of quantity of employees and size of business. Organization employees with 1–9 employees are only 14% in large companies and more than 250 employees (OECD average of 55%) in Mexico. In addition, SMEs have identified some factors such as turnover, number of employees, and total assets [19]. In 2005, the APEC was defined as a small-medium enterprise that has less than 5 employees, providing 30% of private-sector jobs in Taiwan [20] according to the Ministry of Economic Affairs. Micro-enterprises have less than 20 employees, and the service industry has 5 or fewer [21]. Therefore, a micro-enterprise generally refers to a small-scale enterprise that employs fewer than five employees. According to the results of the Census of e-commerce in 2011, there were many industries, as many as 940,000 businesses, and services in less than five people in Taiwan, which accounted for about 79.79% of the total number of companies. The concept of SMEs and micro-enterprise businesses is in many developing countries. The main drivers of economic, employment growth, and poverty reduction of developing countries contain more innovation and new business development. In terms of countries, employment was relatively low or high (between 60% and 90%) in China, Hong Kong, Indonesia, Japan, Korea, Mexico, The Philippines, Taiwan, United States, Vietnam, and the six other countries (Malaysia, New Zealand, Papua New Guinea, Russia, Singapore, and Thailand).

The challenging role of the Internet was analyzed, and it was found that the vast majority of SMEs do not connect due to lack of knowledge about the internet; the other barriers of connection are listed, such as costs, lack of workers, lack of time, and perceived risks [22]. Some reasons suggested for this view are that it is too expensive, and that the scope of the task is too complex and difficult. SMEs are concerned with day-to-day problems and have little scope for adopting a longer-term strategic perspective. The use of the Internet is one of the most important ways in which connectivity can assist SME internationalization. This paper uses a case study approach to participate in the acceptance

of cross-border e-commerce projects to guide five SMEs and micro-enterprises as the research objects, and to investigate the factors that companies consider when choosing a cross-border e-commerce platform [23].

The adoption of sustainability in manufacturing methods has become important for industrial organizations to sustain in their business operations, such as a Green Lean Six Sigma, which not only improves green performance but also increases the financial stability of the industry [24]. The integrated application of tools and techniques from each approach complements a common focus related to sustainability enhancement. The proposed conceptual framework provides systematic guidance from the project selection phase to solution sustainment [25]. The circular economy holds great potential to overcome manufacturing waste and offers competitive solutions with significant impacts on globalization in terms of the product quality, cost, and user experience aspects. These have created a playing field for the manufacturing industry as product life cycles are shortened, resulting in the addition of multiple products to the production line. This results in an increased complexity of input materials, operating costs, and waste generation for the manufacturing system [26], which can reduce conflicts with small and medium-sized enterprises through e-commerce platforms.

### 2.2. Cross-Border E-Commerce

Concerning studies on the effect of cross-border e-commerce factors, Yang et al. [27] empirically studied the impact of marketing on the development of SMEs in Taiwan. Some research has studied cross-border e-commerce, focusing on consumer trading welfare. Meanwhile, Gomez-Herrera et al. [28] also found that trade costs associated with overcoming language barriers doubled because cross-border e-commerce shares the same infrastructure as offline transactions. According to Hanson and Kalyanam's [29] argument, online marketing uses information technology with corporate business activities and business methods to carry out marketing activities, with the goal of marketing. They found the growth of cross-border e-commerce reduced distance-related trade costs by the World Customs Organization (WCO) [30]. The global supply chain model had been a core competency in the international trade market, and appropriate trade associations [31] should promote the platform of public service of international industries. The definition of electronic commerce (E-Commerce) is as follows:

1. E-commerce communication perspective is the transfer of information, products/ services, or payments over the telephone, computer networks, or any other electronic aspect;
2. E-commerce business process perspective is the application of technology towards the automation of business transactions and workflows;
3. E-commerce service perspectives is a tool that addresses the desire of enterprises, consumers, and management to cut service costs while improving the quality of goods and increasing the speed of service delivery;
4. E-commerce online perspective provides the capacity to buy and sell products and information on the internet, as well as other online services [32].

The term indicates that a cross-border e-commerce trading platform refers to a virtual environment in which a different country's trading units develop and negotiate with payment. Yoo et al. [33] argued that electronic payment is also an important part of cross-border e-commerce that can improve transaction convenience.

The European Commission (EC) pointed out that cross-border e-commerce is a seller of online transactions that are not in the country where the buyer lives, and also includes travelers who travel to another country of residence to complete the transaction. Cowgill and Dorobantu [34] investigated the influence of cross-border e-commerce on international trade, using domestic trade and data. The empirical study clarified the role of international services throughout the cross-border e-commerce promotion with factor analysis [35–37]. Therefore, the concept of cross-border e-commerce is expressed as follows: cross-border e-commerce web-based digital transactions, information collection, customs, payment,

services, and other business distribution. To promote e-commerce trading platforms, the development of cross-border e-commerce should improve the information of international trade process containing many enterprises and private businesses aware of the investment highlights in the cross-border platform [38,39].

Cross-border e-commerce includes many elements, which are transaction objects, transaction channels, goods circulation, fund delivery, information, and bill exchanges [36]. Based on these factors, there are various types of cross-border e-commerce features, distinguished according to the nature of the market relationship and technology [40,41]. There are three major types of e-commerce: business to business (B2B), business to consumer (B2C), and consumer-to-consumer e-commerce (C2C). Internet-based B2B e-commerce has two features: net marketplaces and private industrial networks. Net marketplaces bring thousands of buyers and sellers to a single digital marketplace, and many-to-many, as well as one-to-many, relationships. Private industrial networks bring a few strategic business partner companies together and support many-to-one/many-to-few relationships. B2C e-commerce has eight features: popularity, richness, Tran's nationality, openness, interactivity, density, customization, and community provider. According to the Ministry of Economic Affairs, Taiwan's report of "B2C E-Commerce Development Strategies", the industry has three characteristics: a densely populated area, a small domestic market, and the use of C2C access by businesses. The study finds B2C cross-border e-commerce online transactions cross-border e-commerce, and SMEs cross-border e-commerce suppliers. Both are platform providers based on the cross-border platform, Taiwan [42].

Business-to-Business E-commerce (B2B): These are online businesses selling to other businesses. This type of e-business is also known as inter-organizational e-commerce. In this type of e-commerce, businesses link with their suppliers and distributors to exchange documents globally and process payments; it is more efficient and productive [43]. B2B e-commerce helps organizations to decrease the cost of purchase orders, production, and delivery. Additionally, it helps companies to keep track of their documents and inventories, thus reducing inventory restocking time and improving services in the long-term [32]. On the other hand, intra-organizational e-commerce allows for the communication and transfer of documents within the same companies. It is used to enhance communication between managers and their employees through video conferencing and electronic mail.

Business to Consumer E-commerce (B2C): This type of e-commerce is concerned with selling products and services directly to consumers. There are several business models of B2C, the most relevant ones for this study being the case of online retailers [44]. They are online retail stores (retailers) that offer products to customers all over the world. Some retailers are called "clicks and mortar" or "clicks and bricks" if they own a physical store and sell their products and services online. Several other variations of retailers exist, such as online versions of direct catalogs, online malls, and manufacturer-direct online sales [45]. Faraoni et al. [46] classified e-tailing business models according to the type of sites that sell directly to consumers. They include direct marketing sites for manufacturers who sell to consumers, pure-play-e-tai1crs with no physical stores, and the traditional retailers with websites, which are referred to as 'click and mortar' or 'click and bricks'.

Consumer-to-Consumer E-commerce (C2C): This type of e-commerce provides an opportunity for consumers to sell among each other. Consumers prepare the product for the market, and rely on market makers, such as e-Bay, to provide a catalog and search engines to help in finding and purchasing the products [47]. In this type of e-commerce, some individuals use numerous auction websites that permit them to sell their products [48].

### 2.3. E-Commerce Success Model

Based on DeLone and McLean's [49,50] information system successful model for e-commerce, the most six important factors concern information quality, which refers to the information products that produce information quality, including information accuracy, reliability, meaningfulness, and integrity. System quality refers to the information system design itself, including the system's reliability, ease of use, and flexibility, which produces

the information. System usage refers to the user's interaction of the product with the degree of use of the system. User satisfaction refers to the user's satisfaction with recipients, and the decision maker's use of the system. Personal impact refers to the individual influence by-products have on management decisions of the system users, and organizational impact refers to the impact of the system product on the organization's performance.

Sandy [51] proposed the development of an e-commerce model of a comparative study of small and medium-sized enterprises (SMEs). The study demonstrated the promotion of e-commerce between the two countries: Australia and Singapore. The study conducted interviews of small businesses in both countries. Respondents showed that perceptions of e-commerce are positive, and showed the multiple regression analysis to determine the success factors that affect the implementation of e-commerce [52]. The surveys were affected by the overall satisfaction of the regression, which carried out 5 out of the 19 factors. It was found to make a significant contribution in Australia: observability, communication channel, customer pressure, supplier pressure, and governmental support; the survey data used for regression and found only three factors: size of the firm, perceived readiness, and observability (having the necessary organizational tools), which had a significant impact in Singapore. It provides a more comprehensive understanding of the implementation of other factors, which explains the behavior of SMEs; it is being considered when it comes to testing. This decision is very important for Australian SMEs' adoption of communication methods, government support, and external pressure considered from customers and suppliers. The results of this study can help companies understand the relative importance of these factors; SMEs can avoid spending in these less important factors above, the limited resources to achieve the success of cross-border electricity supplier, as shown in Figure 1:

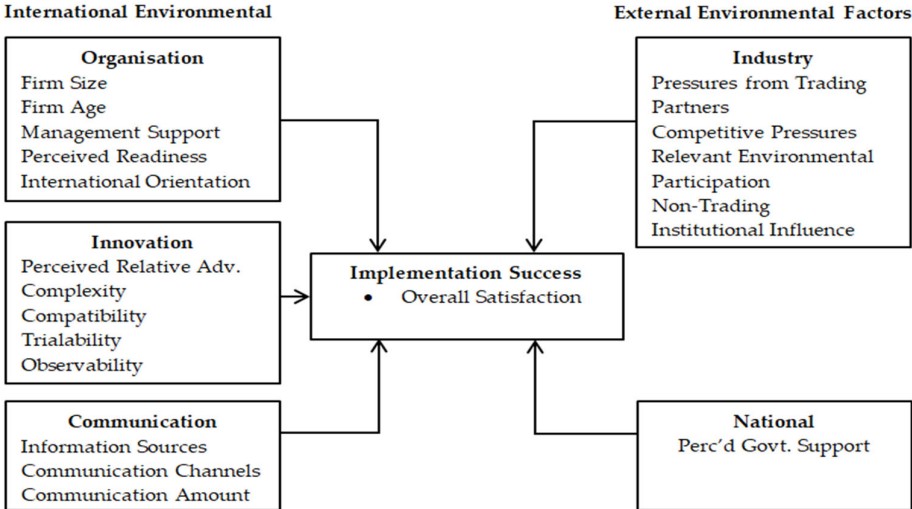

**Figure 1.** Influence factor of implementation success of e-commerce diagrams [51].

Kraemer et al. [53] pointed out the relationship between firm globalization, the scope of e-commerce use, and improved firm performance using a large-scale cross-border market. The study found that globalization leads to both greater scope of e-commerce use and improved performance over the larger-scale, measurement as efficiency, coordination, and market impact. The scope of e-commerce use leads to the greater firm performance of all three types. The results provide support for Porter's [54] argument—that upstream business activities (B2B) are more global while downstream business activities (B2C) are more local or varied (multi-domestic), as shown in Figure 2.

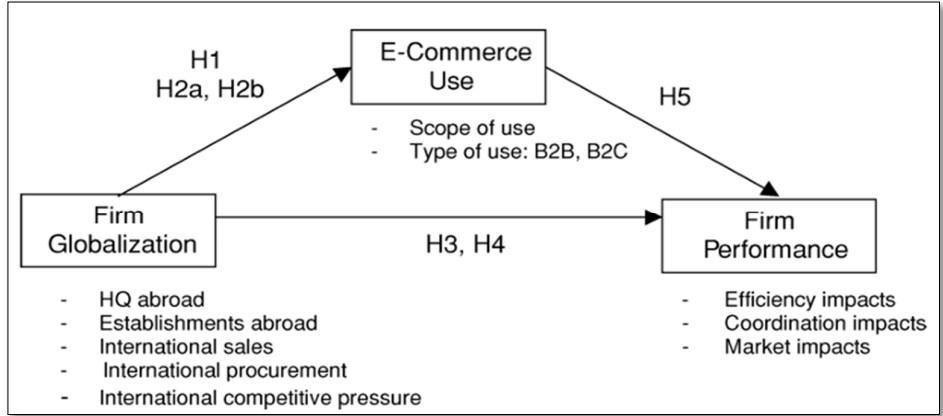

**Figure 2.** Corporate globalization and e-commerce performance [53].

*2.4. Research Framework*

This study explores the factors that influence Taiwanese small, medium-sized, and micro-enterprises in choosing cross-border e-commerce; it compared to analyze the difference between SMEs and micro-enterprises and cross-border e-commerce platforms. The capabilities of the enterprise will influence their consideration of cross-border e-commerce platforms; the capabilities of the SMEs and micro-enterprise divides into three parts: products, knowledge, and marketing. These factors determine the selection of SMEs and micro-enterprises for cross-border e-commerce, which has four factors: economic, social, technological, and legal factors [14]. In the following section, we will discuss the factors in this study. The research model is shown in Figure 3.

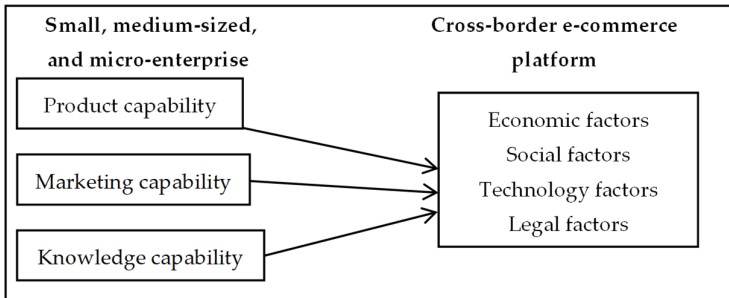

**Figure 3.** Research model.

This study considered the factors which discuss the literature review. According to the following framework set, the hypotheses development of this research is as follows:

**Hypothesis 1 (H1):** *Small, medium-sized, and micro-enterprises will have greater performance when using cross-border e-commerce.*

**Hypothesis 2 (H2):** *Small, medium-sized, and micro-enterprises' product capability will have higher levels of impacts (Economic, Social, Technology, and Legal) by using cross-border e-commerce.*

**Hypothesis 3 (H3):** *Small, medium-sized, and micro-enterprises' marketing capability will have greater performance (Economic, Social, Technology, and Legal) when choosing cross-border e-commerce.*

**Hypothesis 4 (H4):** *Small, medium-sized, and micro-enterprises' knowledge capability will have greater experience cross-border e-commerce use (Economic, Social, Technology, and Legal).*

### 2.4.1. Small, Medium-Sized, and Micro-Enterprises

The capabilities of the SMEs and micro-enterprise divide into three parts: products, knowledge, and marketing. When Armstrong G. [55] gave a speech in Taiwan, he also emphasized that marketing is the key factor of business success. The biggest challenge of business success is more important in marketing; it must know more about the customer than competitors, as after that they can buy products and services [56]. If SMEs and micro-enterprises want to increase, their business should understand and catch up with the current trends and knowledge of cross-border e-commerce [24,25].

Product capability: Products that companies are currently selling on cross-border e-commerce platforms, including the domestic and international certification of relevant products and changing customer needs due to technology performance [25]. The product features are suitable for cross-border e-commerce, whether products have uniqueness, and the competitiveness of product life, etc.

Marketing capability: The company has online marketing tools, including official websites, Facebook, or other social interaction pages, whether there is systematic marketing of these marketing tools, and whether there are strategic plans for future marketing layouts.

Knowledge capability: The company's knowledge of cross-border e-commerce-related knowledge, including the understanding of virtual and real integration, understanding of the trend of cross-border e-commerce operations, and the company will have relevant plans for future.

Table 1 shows the definition and evaluation criteria of the three enterprise capabilities. This analysis evaluated the company's capabilities by high or low standards. The evaluation criteria are through interviews and observations of the company's products, marketing, and knowledge of e-commerce.

**Table 1.** Definition and evaluation standards for enterprise capabilities.

| Capabilities | Definition | Evaluation Criteria |
| --- | --- | --- |
| Product capability | Whether the company's products are related to domestic and international certification, whether product features are suitable for cross-border e-commerce, whether products have uniqueness, and the competitiveness of products in the market, etc. | (1) Whether the product formula has its uniqueness and irreplaceability (2) Whether the product has market competitiveness (3) Whether the product has domestic and foreign certification (4) Whether the product is suitable for cross-border e-commerce |
| Marketing capability | Does the company currently have online marketing tools, including official websites, Facebook, or other social interaction pages; is there a systematic way to operate these marketing tools, and whether it is strategic for future marketing | (1) Whether you are using online marketing tools (2) Whether to understand and invest in these marketing tools (3) Whether to understand the brand positioning of their products (4) Is there any development strategy in marketing in the future? |
| Knowledge capability | Corporate understanding of virtual reality integration, understanding of the current trend of cross-border e-commerce operations, and whether the company has plans for cross-border e-commerce in the future | (1) The degree of understanding of the relevant counseling methods mentioned in the counseling (2) Ideas or plans related to cross-border e-commerce for their own companies |

### 2.4.2. Cross-Border E-Commerce Platform Selection Factors

Bingi et al. [14] categorized the factors that companies need to consider when selecting a cross-border e-commerce platform as being divided into four factors: economic, social, technological, and legal.

Economic Factors: The economic factors refer to the cost justification of projects, and all kinds of users who access the internet, around the issues of infrastructure upgrade and

the skill shortage that is necessary to consider in e-commerce [26,57]. Economic factors span three components, as follows. One, marketing—SMEs must play a role in cross-border e-commerce; they need marketing services such as advertising, website marketing, marketing tools, sellers' goodwill, and the trust between buyers and sellers. Second, pricing related to the profit of the intermediaries (such as platform suppliers, professional sellers) and suppliers, and the price of the consumer's purchase. Therefore, it is very important to make the parties satisfied and make the consumers willing to pay. Third, market analysis—when a supplier wants to sell goods abroad, their opponents will be scattered around the world, so the assistance of market analysis can help suppliers analyze their product advantages in new markets.

Social Factors: The social factors refer to privacy and security on the internet, the challenges of cultural diversity, the questions raised by user resistance and inadequate trust, and the absence of a tactile medium for online sales. Because of the cultural diversity of cross-border e-commerce, it is very important to understand the culture of the place of sale. Social factors refer to the following two factors as follows. One, culture—sealing things in the target market before they integrate into the local market. Then let your consumers accept the product. Second, customer service—when the customer has a problem with the product they will ask the seller for help or further processing the return. At this time, an external communication direction is required. Therefore, this study classifies customer service as the supply. The business has been established by itself and is provided by the platform provider.

Technology Factors: Technology factors refer to security, stability, reliability, communication protocols, bandwidth availability, and integration with existing applications. Technology factors describe two important factors, payment and logistics, as follows. One, payment—when SMEs choose an online shopping platform they will give priority to the website's measures for customer privacy and security, strengthening the protection of consumer data and the security of the payment. Therefore, the current cross-border e-commerce is commonly used. The payment methods are three types of payment: third party payment, credit card, and overseas remittance. Second, logistics—Liu et al. [58] described four types of logistics as postal parcels, international express, cross-border logistics, and overseas storage. Kawa [59] presented the conception of an integrator in cross-border e-commerce to solve the logistics problems of cross-border e-commerce by the whole supply chain. Cross-border e-commerce influences supply chain management to improve logistics systems [60].

Legal Factors: Legal factors refer to intellectual property rights, the legal validity of e-transaction, taxation issues, and the violation of local law. Because of popular demand, customers could learn the following from the internet: security, quality, price, and the service information. Legal factors describe the law of security as follows. One, certification—different countries have different import-related laws and regulations. For example, in China, there are three certifications (business registration certification, tax registration certification, and commodity registration certification), and Islamic countries have halal certification and some regulations. Second, taxation—different countries have different tax rates for product types, and taxation methods are different. Before the 2016 tax reform, China defined cross-border e-commerce retail imports as postal tax, tariffs, and import VAT, and the combination of consumption tax and taxation is different. After tax reforms, the taxation of each country is changing every day.

## 3. Research Methodology

### 3.1. Method

This research used a case study method to review different kinds of examples, and this method provides researchers with a more significant contribution to this research [61]. In addition, the present studies focus on case study, which is suitable to examine the significant research of their study [62]. Multiple case study methods are used by authors researching the aspects that affect small, medium, and micro-enterprises as they examine the CBE.

These methods provide a deeper knowledge of the selected example and help to clarify the theories [63,64]. The growing features of this research considers the multiple case study methods as more skillful when the suggestions are more deeply explained in common empirical evidence. Furthermore, multiple cases study give allowance to the researcher to explore research objectives and theoretical progression [65]. While the multiple case study method gives more powerful and authentic evidence than a single case study, the previous is more precious and essential to put in an application [66].

Given the incomparable outcomes, the case study technique's benefits have been scrutinized for their lack of meaningful conclusions. In particular, scholars have concurred that the case study method is inappropriate for hypothesis testing [67]. This point of view is not accurate, as the case studies give the researcher a chance to grow new hypotheses [62]. When studying a small number of cases, it is very difficult for the researcher to explore and criticize the statistics, however, logical conclusions based on simple descriptions can be explored [68].

### 3.2. Sample and Purpose

The sample for this study aimed to find out the influence factors of CBE, and it was conducted among the enterprise experts and managements in Taiwan. The structure of the interview questions was developed, and this study focuses on discussions with company experts. The case study of Yin [69] also requires further gathering of evidence through interviews and direct observations. There are 10 interviewees in different companies, 5 small and medium-sized enterprises, and 5 micro-enterprises for the cross-border e-commerce counseling. These companies belong to different areas in Taiwan and provide various types of products and service. This study was used to collect data by interviewing small- to medium-sized enterprises and micro-enterprises with a good relationship to their companies' management, such as companies' manager, deputy general manager, general manager, CEO, principle, company representatives, and special assistance [7–9]. The interviewee respondents of this study were from small, medium-sized, and micro-enterprises. Basic information is displayed in Table 2.

**Table 2.** Interviewee information.

| No. | Company Name | Job Title | Major Merchandise |
|:---:|:---:|:---:|:---:|
| 1 | SME A | Deputy general manager | Fermented bean curd |
| 2 | SME B | Manager | Loose fish, processed fishery |
| 3 | SME C | General manager | Nano soap |
| 4 | SME D | CEO | Floss, processed meat products |
| 5 | SME E | Manager | Footwear |
| 6 | Micro-enterprise A | Principals | Essential oil |
| 7 | Micro-enterprise B | Manager | cosmetic |
| 8 | Micro-enterprise C | Company representatives | Soap film, handmade soap |
| 9 | Micro-enterprise D | Special assistance | Coffee, tea, and other drinks |
| 10 | Micro-enterprise E | Principal | Ice cream, shortbread |

Organization SME A was established in 1961. This company is located in the central area of Taiwan, and started as a small fermented bean curd factory. After years of wholeheartedly operating "Inheriting of the ancient flavor of fermented bean curd", it has become loved by locals and villagers and gained a good reputation until it was handed over, and the second generation of succession management created the company SME A. Moreover, the second generation focuses on innovative flavor development and process improvement and expanding a larger customer base market. So far, the third generation has taken over the innovative future development of the brand and overseas marketing.

In addition, the owners mentioned that they have physical channels in a certain market in Taiwan, under the affirmation of retail stores and chain supermarkets. The business map has continued to expand. Product research and development have been continuously updated with the trend. In terms of virtual channels, there are GoHappy, retail stores, and self-built official websites in China [10]. Through these channels, a certain fixed customer has been accumulated in China. In foreign countries, there are dealers or supermarket chains that are selling on behalf of them. However, the business owners still want to invest in the development of cross-border e-commerce. The business owners believe that their fermented bean curd is limited due to product features. Many people in Western countries do not understand this product. Even in Eastern countries, only certain regions know of the fermented bean curd product. The business owners want to open up the popularity and acceptance of the product through the boom of cross-border e-commerce, then put it on the physical channels in foreign countries.

Organization SME B was established in 1950 and has operated for more than 60 years. It is located in northern Taiwan, and this city's most famous product is swordfish pine. In this city, there is a swordfish pine specialty shop in the market. In 1984, in order to meet the demand, the owner established a modern and automated health and safety factory in the Dali industrial zone, Taichung. To improve the quantity and quality of products to a higher level, they are committed to combining fish production with agriculture to create a variety of products so that more people can eat this delicacy. During the counseling, it was found that SME B has been operating for a long period of operation. There is a fixed number of customers on the physical channel. However, the owner also mentioned that they have been working hard on the virtual channel for a period of time. If you have your own official website, go to various domestic e-commerce platforms, and continue to sell online, but the sales volume is less than one-tenth. In the cross-border e-commerce sector, SME B has already sold in mainland China [5,10]; the problem encountered by the owners is that if mainland China needs three certificates, they will prepare three certificates. However, each country has different requirements. The biggest problem for owners at present is that they do not understand the requirements of various countries in terms of cross-border e-commerce, and they are in a dilemma of not knowing where to ask for help.

Organization SME C was favored by Chung Hsing University of the business of philosophy in 2007. After that, the chance to work with Chung Hsing University opened up, and participants were involved in significant initiatives, including academic patent technology transfer collaboration, and vowed to build unique goods. Therefore, products with special patents like nano-silicon soap were born. After years of hard work, the first store was established in 2014. Moreover, the owner mentioned that SME C had no storefront when it was founded, and it started from online sales and word of mouth. So far, virtual channels still account for 60% of total revenue. The owner mentioned that in the future, he hopes to put the resources on the virtual channel, and only do the shipment.

Organization SME D was established at the first Rousong store in Taiwan, 1966. After decades of development, it is now an old store with multiple stores and sales channels. The first generation of owners adhered to the principle of fresh raw materials being fried every day during the start-up period. This has enabled SME D to make good achievements in the country in the past 40 years. Now they not only take root in Taiwan, but also want to cross the sea to develop across the sea so that more people can eat delicious delicacies full of happiness. During the counseling, it was found that the company has gradually been handed over to the second generation to operate, and it is in the stage of transformation. Because SME D has been a time-honored brand for more than 40 years, when it is stimulated by young employees' new ideas, it is inevitable that there will be disputes and disagreements. In addition, during the transition period, the owners wanted to vigorously develop and expand overseas businesses and tried various methods during this period. However, due to product characteristics, the sales have not met expectations in many ways, and the owners are currently exploring a cross-border channel suitable for their own enterprises.

Organization SME E was established in inheriting 40 years of craftsmanship and it is a socks and shoe brand enterprise. In order to provide comfortable and stable shoes for growing children, they developed a non-slip rubber sole with a stable back heel, and a non-binding sock body that allows the toes to move in a wide shoe shape. For activities, yoga insoles with excellent cushioning and an exclusive patented safety shoelace design have been selling wildly in major channels in just a few years on the market. In addition, the owner mentioned that SME E has developed in both physical and virtual channels. The performance ratio is about half, and the domestic side has accumulated a certain amount of strength. In terms of internationalization, they are also actively participating in exhibitions, or put on shelves on major foreign platforms. However, the owners still find it difficult to shine in the international arena.

Organization micro-enterprise A is located in Taiwan, and it was established 16 years ago [17]. An essential oil company has spanned 16 years of cross-border e-commerce. In the past 16 years, it has done everything that should be done in cross-border e-commerce, such as online marketing, Facebook marketing, word-of-mouth marketing, platforming, etc. During the interview, the owner mentioned that developed physical and virtual stores have been accumulating a group of customers. However, in the face of fierce international competition, they find that if they no longer adjust their pace, they will be eliminated by this society. Now is the time for transformation. At the beginning of this year, micro-enterprise A put away all the channels in Taiwan and stopped selling for a few months. In the past few months, they have targeted the U.S. market. The first step to enter the U.S. market is that they have to get FDA certification for this type of product [11,18]. After that, they continue to develop new products and send them for verification. In the end, it successfully obtained the certification and opened up its popularity in the United States, and because of this, it was sold to Japan and mainland China. The owner firmly believes that the brand of the product is the king of sales, and cross-border e-commerce is just one of the relay tools to enhance the brand and increase the number of points of sale.

Organization micro-enterprise B was established in the 100th year of the Republic of China [5]. It mainly does the design, research and development, and purchasing of raw materials for beauty products. The production part is subcontracted, and the owner insists that the customer feels at ease using products that are truly harmless to the skin. Products with high-efficiency ingredients provide customers with intimate product-consulting services and new knowledge of maintenance. In addition, micro-enterprise B believes that their product strength has met international challenges. However, cross-border e-commerce often depends on marketing and product descriptions to determine consumers' buying behavior. It just so happens that they lack marketing resources because they are micro-enterprises. Therefore, the owner stated that overcoming this problem and letting those who have never used them know that the benefits of their products are the problems that should be solved at present.

Organization micro-enterprise C was established in 1998, and it is a handmade soap raw material industry. In particular, it cooperates with American technology partners to create a trend of domestic DIY handmade soap and artistic candle styles which are different from the monotonous shape of ordinary traditional handmade soaps. Micro-enterprise C has successfully integrated molding and soap-making technology into more local characteristics and special raw materials while constantly creating the possibility of infinite development of handmade soap. During the counseling, micro-enterprise C owner mentioned that they currently do not have physical channels, and all sales come from virtual channels. The owner said that he has tried to go hand in hand with virtual and real channels before, but at present, he believes that virtual channels are the bulk of development. Firstly, it can save costs, and secondly, the customer return rate is stable and the virtual channel is sufficient for the purchase process. However, the current owner is also encountering marketing problems. Although the micro-enterprise C has spread across many countries, due to limited resources, they rely on passively waiting for customers to

come to their door. There is no marketing plan yet, and they hope to make breakthroughs in this area in the future.

Organization micro-enterprise D was established in 2012, and mainly focuses on coffee, tea, and dessert-related products. Special consideration has been given to the selection of raw materials, manufacturing process, and packaging. Firstly, represent innovation, from product innovation to service innovation, aiming to give customers a new feeling; Second, represent the two concepts of "safety" and "happy". Let customers drink with peace of mind and happiness. Moreover, the owner mentioned that there are currently two physical channels. One is mainly based on natural beverages, and the target customer group is students. The other one mainly focuses on specialty coffee, and the target customer group is office workers. However, the virtual channel mainly sells gift box boutiques. At present, the owners think that their brand is not big enough, so they still focus on physical channels.

Organization micro-enterprise E was established in 2010. At first, it was just a small vendor selling ice cream on the Iron Horse Road in Taiwan. Then, in November, it was voted third place in the "Greater Taichung Campus Food". In 2011, it became famous with "100th Anniversary of the Founding of the People's Republic of China Ice Cream" and "Summer Love Ice Cream". In order to match these virtual markets, such as group buying and home delivery, the owner developed and patented "pure milk shortbread", and started to set up counters in chain retail stores in 2013.

### 3.3. Data Collection Processes

In this research we used multiple cases study methods, as this method provided a better opportunity for wider explanations. A case study is the best way to explore cases, and to explain them and make them understandable [70]. While multiple case study methods provide stronger evidence and more truthful results than single case studies, prior to this, more precious essentials take time to apply [71]. The data collection method is a primary data collection method. The primary data collection methods are interviews, case studies, questionnaire methods, and others. Interview research methods were used to collect relevant information. The first part collects the basic information of respondents to confirm relevant interview questions and answers. The second part of the interview outlines uses semi-structured interview questionnaires for marketing, logistics, operations, etc. On the other hand, interview respondents were asked to respond according to their own company's status. According to the definition of the case study data proposed by Yin [69], the secondary data of this study are as follows: the secondary data content is personal career support counseling, planning to promote the integration of micro-enterprise networks into internationalization, and written information such as corporate report-related innovation can help this research to identify the key variables. In this study relevant information was collected from September to October 2016, which verifies the data without errors.

## 4. Results and Discussion

### 4.1. Comparison between SMEs and Micro-Enterprise Based on Three Capabilities

Based on the case analysis, the study compared and analyzed the high and low results of the three capabilities, "product", "marketing", and "knowledge", to obtain five different categories to analyze SMEs and micro-enterprises. Similar to [72], in this study, cross-border e-commerce platforms can facilitate trade in three key ways. Digital e-commerce platforms can serve as enablers through which new businesses can be created and traditional ones can be transformed. This study identifies differences in motivational influence between the two types of entrepreneurs, which translates into different financial outcomes. Although the business strategies based on the e-commerce platform are similar between the two types of entrepreneurs, the difference between the survival and growth motivation can be clearly seen from the difference in the business expansion speed of the two groups of entrepreneurs.

The findings differ from the conventional view that entrepreneurs have limited access to resources (financial, human, social) and are therefore excluded from actively participating

in transnational activities. The digital platforms allow necessity entrepreneurs to conduct business in exactly the same way as opportunity entrepreneurs. The characteristics of cross-border e-commerce are between SMEs and micro-enterprises, as shown in Table 3.

**Table 3.** Summary of SMEs and micro-enterprises' capabilities.

| Enterprise Name | Product Capability | Marketing Capability | Knowledge Capability | Main Products |
|---|---|---|---|---|
| SMEs A | low | high | high | Fermented bean curd |
| SMEs B | high | low | high | Dried fish floss, processed fishery |
| SMEs C | high | high | high | Nano soap |
| SMEs D | low | high | high | Floss, processed meat products |
| SMEs E | high | high | low | Footwear |
| Micro-enterprises A | high | high | high | Essential oil |
| Micro-enterprises B | high | low | high | Cosmetic |
| Micro-enterprises C | high | low | high | Soap film, handmade soap |
| Micro-enterprises D | low | low | low | Coffee, tea, and other drinks |
| Micro-enterprises E | high | high | low | Ice cream, shortbread |

4.1.1. The Study Compared and Analyzed the High and Low Results of the Three Capabilities, "Product", "Marketing", and "Knowledge", to Obtain and Analyze organization SME A

Product capability—Low: Table 3 indicates that the enterprise abilities of product capability are low. Because the product quality has passed various food certifications, there are also groups of long-standing consumers in terms of taste. Nevertheless, in the face of cross-border e-commerce, their products are restricted everywhere because of their characteristics. The packaging of glass bottles is easily broken, and it is difficult to transport over long distances. Consumers in the place where the taste is sold may not be able to accept such a novel product. This is where they need to reconsider the product.

Marketing capability—high: Because the organization has significantly succeeded by the second and third generations. The successor of the third generation has studied the relevant marketing methods. In the interview, he mentioned the importance of the brand. He believes that in the fermented bean curd industry the generation of the work is not a long-term solution. Therefore, in the future, he plans to use cross-border e-commerce to open up international popularity and apply different marketing strategies in different regions. Let a part of the customer group accept it first, and then hit the physical channel market.

Knowledge capability—high: Because the enterprises have a certain understanding of cross-border e-commerce-related knowledge through the distributors responsible for international sales. The actual actions of cross-border e-commerce in the future are also underway. There are currently negotiations with Singapore and Malaysia platform vendors.

4.1.2. The Study Compared and Analyzed the High and Low Results of the Three Capabilities, "Product", "Marketing", and "Knowledge", to Obtain and Analyze Organization SMEs B

Product capability—high: Because SMEs B has its own factory, it can provide extremely customized services, and the products can constantly change in response to customer needs in various places, maintaining the concept of product diversification and having relatively low-priced products. There are also exquisite products with high unit prices, and with extremely high plasticity.

Marketing capability—low: In terms of marketing capabilities, SMEs B has a fixed customer base because it started form physical channels such as the market. However, it belongs to a traditional industry and faces the competition of the big environment. There is a tendency to lose customers, and it is currently in a transitional stage. In terms of transformation, the marketing plan is still being drafted, and it will take some time to digest and determine the target.

Knowledge capability—high: About the knowledge capability, the owner talked a lot about the advantages and disadvantages of the company's products in various cross-border e-commerce platforms, and clearly understood the positioning of their products in the international market.

### 4.1.3. About the High and Low Results of the Three Capabilities, "Product", "Marketing", and "Knowledge", to Obtain and Analyze Organization SMEs C

Product capability—high: The products of SMEs C have development patents, which are relatively low in substitution, and the packaging is specially designed according to the needs of each customer needs. The nature of the product is biased towards the public, and it is suitable for cross-border e-commerce in terms of customer acceptance and transportation.

Marketing capability—high: SME C has a professional marketing planning team that is responsible for product promotion. There are also many requirements in terms of brands; it hopes to create market value through the particularity and advantages of products. It is better to take back retail dealers who have different quality control, and also to defend their own brand value.

Knowledge capability—high: SME C knows the direction of their products in the future international e-commerce market and has been implementing plans systematically. They hope to accumulate customer traffic through the boom of cross-border e-commerce.

### 4.1.4. About the High and Low Results of the Three Capabilities, "Product", "Marketing", and "Knowledge", to Obtain and Analyze Organization SMEs D

Product capability—low: After more than 40 years of continuous research and development, the products have a certain quality. However, it is limited to meat products and Taiwan is a foot-and-mouth disease epidemic area, which makes it very difficult to export. Moreover, the owners feel that eating meat floss is not as indispensable a food in the home as before. The owners are currently encountering this major problem.

Marketing capability—high: In terms of marketing, SMEs D is committed to studying the consumption habits of today's customers and wants to transform traditional food into a new style to enter the young market. The owner's marketing in the domestic market such as direct-sale stores, department store counters, and airport stations as the first front line where there are many tourists. Although online marketing and word-of-mouth marketing are also in parallel, so far, the effect of these physical stores have achieved better results. However, the owner believes that to enter the international market in the future, online marketing must be combined with local trends in order to make their products be accepted by the public abroad.

Knowledge capability—high: Owners have been in consistent contact with cross-border e-commerce. At present, the mainland and South Korea are the main sales locations in the trend of cross-border e-commerce. Before they do cross-border e-commerce, they have to send people to the field to observe the station and evaluate whether the product can stand in this area. After a period of planning and evaluation, they may choose to release the products.

### 4.1.5. The Study Analyzed the High and Low Results of the Three Capabilities, "Product", "Marketing", and "Knowledge", to Obtain the Organization SMEs E

Product capability—high: The products of SMEs E are aimed at the children's footwear market. They are different from other products on the market in terms of uniqueness. They developed their own webbing patents. In addition, owners believe that their products can have advantages in cross-border e-commerce, as the same type of socks and shoes are cheaper than many places in terms of cost.

Marketing capability—high: At present, owners have adopted virtual and real integration methods, using virtual marketing methods such as fan groups, Line@, etc., to attract customers and bring customer traffic to physical stores. Currently, it is also available on major domestic platforms because of the product characteristics. They have accumulated a certain amount of popularity on the Internet.

Knowledge capability—low: At present, the owner only collects relevant cross-border e-commerce knowledge by participating in foreign exhibitions. There is currently no clear goal in cross-border e-commerce.

### 4.1.6. The Study Analyzed the High and Low Results of the Three Capabilities, "Product", "Marketing", and "Knowledge", to Obtain the Organization Micro-Enterprises A

Product capability—high: Micro-enterprise A products have been certified by the US FDA and are the first certification in this category. The market has very low substitution, so the product strength is very high.

Marketing capability—high: Micro-enterprise A understands the importance of product brand value. Therefore, the owner has been mainly focusing on foreign markets and obtaining certifications in response to the needs of foreign markets. The world map of products starts from America, then sells to Japan, and then sells to mainland China. It is believed that the needs of the main target group can be used to maximize the brand value. Hence, this kind of marketing method is common in non-micro-enterprises.

Knowledge capability—high: Micro-enterprise A has entered this field since the beginning of global cross-border e-commerce. After years of sales, it has accumulated a certain number of customers. The owner not only has a very good understanding of cross-border e-commerce, but also clearly knows what kind of cross-border e-commerce assistance is needed for the products he owns at this stage.

### 4.1.7. The study Analyzed the High and Low Results of the Three Variables, "Product", "Marketing", and "Knowledge" to Obtain the Organization Micro-Enterprises B

Product capability—high: Because micro-enterprise B is a micro-enterprise, its human resources are very limited, which leads to limited progress in marketing. Although there is already a certain degree of exposure on the Internet depending on the product strength and the real experience of consumers, these so-called exposures are there. Under the premise of having used it, but in the face of cross-border e-commerce, the unique features of the product alone cannot be attractive.

Marketing capability—low: Micro-enterprise B is a micro-enterprise with very limited human resources, which leads to limited progress in marketing. Although relying on the strength of the product and the real experience of consumers, there has been a certain degree of exposure on the Internet, these so-called exposures, based on the premise that they have used it. However, in the face of cross-border e-commerce, it is impossible to attract people just by having unique products.

Knowledge capability—high: Micro-enterprise B's current main direction is to move towards internationalization. The owner believes that cross-border e-commerce is a very effective method. He is very aware of what his products lack most internationally. He wants to use the advantages of products to add value to his brand through exposure in cross-border e-commerce.

### 4.1.8. The Study Analyzed the High and Low Results of the Three Capabilities, "Product", "Marketing", and "Knowledge", to Obtain the Organization Micro-Enterprises C

Product capability—high: The products of micro-enterprise C are very distinctive. In addition to the effects of the same type on the market, he also integrates cultural creativity and entertainment into his products, such as artistic soap, clay soap, and silicone soap mold, to make his products more sophisticated. In order to cater to a more diverse customer base, their products are constantly innovating, and there are more than 1000 products.

Marketing capability—low: Micro-enterprise C is very sophisticated and distinctive in the product part, but the owner said that the current predicament is the problem of marketing. They only put their products on the platform and the Internet, and passively wait for customers to come to their door. Sales are maintained through customer return and word of mouth.

Knowledge capability—high: Micro-enterprise C is very good at cross-border e-commerce. The foreign platforms currently being sold on include Ebay, Etsy, and Amazon,

and spans more than 35 countries. Therefore, they have a good understanding of cross-border e-commerce knowledge.

### 4.1.9. The Study Analyzed the High and Low Results of the Three Capabilities, "Product", "Marketing", and "Knowledge", to Obtain the Organization Micro-Enterprises D

Product capability—low: There are differences in products due to different physical channels and virtual channels. However, on the market, especially internationally, there are too many boutique gift boxes of the same type. The uniqueness of the market is slightly lower. If you want to make gains in cross-border e-commerce, it definitely needs some additional value-added products.

Marketing capability—low: Micro-enterprise D is still in the start-up stage. The exposure of the brand only depends on the promotion of physical channel, and then brings the physical customer source to the virtual channel. In addition, the owner said that the benefits brought by the virtual channel are still too low, so it is impossible to allocate too many resources to the marketing.

Knowledge capability—low: The owner currently wants to try to promote the product internationally, but he does not know much about cross-border e-commerce. Although there is contact, it just keeps the mentality that everyone else is doing it, so try to do it and see it, and then choose your future plans based on your own situation.

### 4.1.10. The Study Compared and Analyzed the High and Low Results of the Three Capabilities, "Product", "Marketing", and "Knowledge", to Obtain and Analyze Organization Micro-Enterprise E

Product capability—high: Micro-enterprise E products have very high market exclusivity because of their patented technology. Although there are many homogeneous products on the market, this patent has achieved something that homogeneous products cannot do. Additionally, the owner has a design in response to virtual channels. The products that are suitable for sale in virtual channels, such as gift boxes and souvenirs, prove that the products of micro-enterprise E are very competitive.

Marketing capability—high: Micro-enterprise E started out as a street vendor. Through participating in marketing selections such as campuses and night markets, the company has become bigger and bigger. The virtual marketing part has its own fan group, official website, YouTube audio and video promotion, and blogger word-of-mouth promotion. Domestically, it has accumulated a certain reputation, and the owner hopes that through this popularity and the assistance of the government, it will be in line with international standards, making the brand bigger and causing a boom in foreign countries.

Knowledge capability—low: Although the owner currently has an understanding of cross-border e-commerce, there is no clear plan yet. It is still impossible to determine which method is the best way to sell his products abroad. In addition, only the owner has access to this type of information, and the rest of the employees have not received this training.

### 4.2. Consideration Factors of SMEs and Micro-Enterprise

Table 4 indicates that two or more repetitive considerations from the two groups of SMEs and micro-enterprises draw as standard references.

The results of Table 4 are further organized to answer the research questions of this study. SMEs and micro-enterprises need to consider those factors when choosing a cross-border e-commerce platform. In this study, the classification of characteristics was summarized, and its context was observed. Through the selection of the five types of enterprises and characteristics of cross-border e-commerce platforms, the differences between SMEs and micro-enterprises are analyzed and compare with considerations factors. According to the capability characteristics of various SMEs and micro-enterprises, this study is based on the relevant data obtained from the research structure and case interviews. This research explores the "product capability", "marketing capability", and "knowledge capability" of the case SMEs and micro-enterprises. To discuss the research questions that SMEs and micro-enterprises need to consider when choosing a cross-border e-commerce

platform, we selected five types of enterprise characteristics of cross-border e-commerce platforms. Moreover, Table 4 results are summarized into five types of consideration factors.

**Table 4.** Considerations for SMEs and micro-enterprises.

| Enterprises Category | Economic Factor | Social Factor | Technological Factor | Legal Factor |
|---|---|---|---|---|
| SMEs A | | Culture | Payment, Logistics | Certification |
| SMEs B | Marketing | Culture | Payment | Certification |
| SMEs C | Marketing, pricing | | Logistics | Certification |
| SMEs D | | Culture | Payment, Logistics | Certification |
| SMEs E | Marketing | | Payment | Certification |
| Micro-enterprises A | Marketing, pricing, market analysis assistance | Culture | | Certification |
| Micro-enterprises B | Marketing, market analysis assistance | Culture | Payment, Logistics | Certification |
| Micro-enterprises C | Marketing, pricing, market analysis assistance | Culture | Logistics | Certification |
| Micro-enterprises D | Marketing | Customer service | Payment, Logistics | |
| Micro-enterprises E | Marketing | Customer service | Payment | Certification |

### 4.2.1. Product Capability

(i)    Consideration of factor analysis

For SMEs A and SMEs D, when choosing cross-border e-commerce, the considerations factors are product capability—low, market capability—high, and knowledge capability—high.

The solutions of cross-border e-commerce platforms of their consideration factors of SME A and SME D considered culture, payment, logistics, and certification. Enterprises have no economic considerations, because the marketing capability and knowledge capability of the company sufficiently deal with economic issues such as cross-border marketing, product pricing, and market analysis. However, in terms of social, technological, and legal aspects, due to the particularity of the product, it is necessary to consider. In terms of culture, this type of product does not necessarily meet the needs of the majority of local people. Therefore, there are many obstacles in marketing and logistics. Due to the difficulty of the product, it is necessary to strengthen the packaging design, supporting measures, and certification. It also takes more time and money to carry out international food certification. The payment process has not been limited by the product characteristics, but the respondents indicated that the variety of payment systems is too complicated. We hope that there will be a window for unified teaching or counseling.

(ii)    There are no differences of considerations between SMEs and micro-enterprises.

### 4.2.2. Marketing Capability

(i)    Consideration of factor analysis

SME B, micro-enterprises B, micro-enterprises C when choosing cross-border e-commerce considerations factors are product capability—high, market capability—low, and knowledge capability—high.

The solutions of cross-border e-commerce platform of their considerations factors of SMEs B considered marketing, culture, payment, and certification; micro-enterprises B considered marketing, market analysis assistance, culture, payment, logistics, and certification; and micro-enterprises C considered factors such as marketing, pricing, market analysis assistance, culture, and logistics. As a result, the product aspect is highly sought by customers. However, due to the lack of investment in enterprise resources on the marketing aspects, the first consideration is the economic considerations of selecting cross-border

e-commerce considerations. The cultural factor is also a lack of marketing aspects of considerations. There is no real understanding of local cultural management, and it is hoped that there would be a window or platform provider to help to solve this type of problem.

(ii)    Differences in considerations between SMEs and micro-enterprises.

In terms of market analysis of the economic capability, both micro-enterprise B and micro-enterprise C indicated that there was no way to handle this matter due to worker limitation and hoped that the platform could assist. In terms of pricing, micro-enterprise C stated that there would be counterfeiters on the platform. Some platform counterfeiters are larger and faster, which has deeply affected their pricing. Micro-enterprise C is a company with limited resources that cannot afford the test of counterfeiting, because counterfeiters often make 80% similar products, but the price is low. The response measures continue to develop new products, but the cost of developing new products exceeds profitability. Therefore, it is very important to consider this factor in selecting cross-border e-commerce platforms. In logistics, for micro-enterprise B and micro-enterprise C, the platform provider must provide the established model that can be directly applied. In contrast, SME B has a problem with the payment. This factor needs to be considered when selecting the platform vendor. There is no difference between the social factors and the legal side.

### 4.2.3. Knowledge Capability

(i)    Consideration of factor analysis

SMEs E and micro-enterprise E, when choosing cross-border e-commerce considerations factors, are product capability and marketing capability—high and knowledge capability—low. The choice of cross-border e-commerce platform focuses on marketing, payment, and certification. Macro-enterprise E considered factors such as marketing, customer service, payment, and certification. Although the companies' domestic products and marketing experience are high they do not understand transnational e-commerce, so they need to first understand the rules of international e-commerce and understand how to sell their products. The payment can only platform businesses, hoping that an area only uses a payment, and saving payment management costs in the legal aspects of the certification. The two owners are not familiar with cross-border e-commerce regulations related to the application. SMEs E hoped someone can provide him or her with applications for the three-license clearance. Micro-enterprise E hoped that someone would provide them with the opportunity to apply for halal certification policy and international food-related inspections.

(ii)    Differences in considerations between SMEs and micro-enterprises

In social customer service, micro-enterprise E believes that they are micro-manufacturers who lack human resources. While the boss of the store often handles customer service, they start to conduct international cross-border e-commerce. In terms of business, customer service will become more extensive and in-depth. Non-workers or those who have not received professional training can afford it. Therefore, micro-enterprise E also considers customer service as one of the considerations, along with economics, technology, and law. There is no difference in legal aspects.

### 4.2.4. Cross-Border Potential Capability

(i)    Consideration of factor analysis

SMEs C and micro-enterprise A are high product aspect, high market aspect, and high knowledge aspect. The choice of cross-border e-commerce platforms focuses on marketing, pricing, logistics, and certification, and the companies focus on the establishment of brands. Micro-enterprises A considered factors such as marketing, pricing, market analysis assistance, culture, and certification. On the economic factors of marketing, this type of business owner believes that a product must sell on an international platform. Passenger traffic is very important, and the operation on marketing is the most important

consideration. Pricing is a consideration, but they are not allowed on the shelves. The platform arbitrarily pulls down product prices. In the legal aspects of certification, and in the process of establishing international brands, it can assist in the establishment of certifications. In addition to the general customs clearance of certification, both companies have obtained international products. Certification, making the product more competitive, enhances international customer acceptance.

(ii) Differences in considerations between SMEs and micro-enterprises

In the economic aspect, micro-enterprise A has similar software to assist, taking into account the worker constraints and the need for a more professional approach to local analysis. In the cultural aspects, SMEs C has sufficient human and resources specialists to local information was collected, but micro-enterprise A had limited resources because they believed that there was a need for platform providers to collaborate with local culture to give product recommendations. In terms of logistics, small and medium-sized enterprises (SMEs) usually ship a lot, because they expressed the hope that platform traders would fully handle the matter. Micro-enterprise A says that as long as the price is reasonable, it can meet the customer's requirements and respect the customer. The legal aspect is no different.

### 4.2.5. Cross-Border Start-Up Capability

(i) Consideration of factor analysis

Micro-enterprise D has low product aspects, low market aspects, and low knowledge aspects. The choice of cross-border e-commerce platform focuses on marketing, customer service, payment, and logistics. Micro-enterprise D has been established for less than five years, so the business is still in the starting stages. In terms of marketing in the economy, e-commerce products have launched. Products are also low due to the same type of products on the market. In addition, the company's initial marketing is also limited, without investing resources; we are currently exploring the marketing knowledge of choosing a transnational e-commerce platform, and we need to understand the information. For customer service in the social sector, the current volume is not high, so the business owner handles it. If the product reaches an international level, the quantity will increase. However, when the customers' demand becomes large, someone will be needed to assist in processing. Due to the lack of multinational e-commerce knowledge, the owner does not understand which kind of payment and logistics to choose and needs the platform provider and the government to help with counseling.

(ii) There are no differences in considerations between SMEs and micro-enterprises.

The result of hypotheses shows, as Tables 3 and 4 indicate, the result of consideration factors as shown in Table 5, and it analyzes and compares the difference between SMEs and micro-enterprises by consideration factors.

**Table 5.** Hypotheses test.

| Hypotheses | Outcome |
|---|---|
| H1: Small, medium-sized, and micro-enterprises will have greater performance when using cross-border e-commerce | Supported |
| H2: Small, medium-sized, and micro-enterprises' product capability will have higher levels of impacts (Economic, Social, Technology, and Legal) when using cross-border e-commerce | Supported |
| H3: Small, medium-sized, and micro-enterprises' marketing capability will have greater performance (Economic, Social, Technology, and Legal) when choosing cross-border e-commerce | Supported |
| H4: Small, medium-sized, and micro-enterprises' knowledge will have greater experience in cross-border e-commerce use (Economic, Social, Technology, and Legal). | Supported |

## 5. Conclusions and Recommendations

*5.1. Conclusions*

This study uses case studies to study the small and medium-sized enterprises and microenterprises who participated in cross-border e-commerce project counseling, a total of 10 companies, and the secondary data analysis, and for different enterprises under the characteristics of enterprises in the economic, social, logistics, and legal, these considerations on the choice of cross-border e-commerce platform. The study found that different enterprise capacity characteristics will affect the choice of cross-border e-commerce platforms. This study summarizes five different types of enterprises: product capability, marketing capability, cross-border potential capability, knowledge-based capability, and cross-border start-up capability. These five characteristics also have different types of options to consider, of which SMEs pay more attention to marketing, culture, payment, logistics, and certification. In addition to the theoretical implications of this study, small and medium enterprises and micro-enterprises in the choice of cross-border e-commerce platform management practice recommendations and proposed future research recommendations. According to the research results, various cross-border e-commerce platforms will provide different business models for enterprises, but these models are not suitable for all enterprises. This study mainly allows business owners to review their corporate capabilities and their corresponding considerations and allow them to select platforms before doing a full range of thinking, then choose the platform that best meets your company.

*5.2. Management Implications and Research Contributions*

This study was conducted through the interviews of 5 SMEs and 5 micro-enterprises and summarized them into five types of cross-border e-commerce platforms: product capability, marketing capability, cross-border potential capability, knowledge capability, and cross-border start-up capability. The situation will help companies in the development of cross-border e-commerce.

In terms of business owners, this study provides five types of reference for enterprises, so that companies can find the most suitable model of cross-border e-commerce platform, and then solve the factors that enterprises consider when choosing a cross-border e-commerce platform. The platform providers can fully understand why companies do not use platforms and learn that the reasons can be adjusted so that platform operators can remain flexible. On the one hand, they retain the original platform functions; they can design a simple cross-border e-commerce platform model for small and medium-sized enterprises and micro-enterprises. In terms of government guidance, the government can assist companies in conducting health inspections when companies participate in cross-border e-commerce counseling cases.

*5.3. Research Limitations*

The research method limitations are explored in this study with consideration factors of models, which is the most influential factor in businesses using cross-border e-commerce.

(i)     Due to the sampling, this study did not specifically mention the industry of the enterprise. However, in the course of the research, it was discovered that several aspects of international e-commerce, such tax legislation, would have distinct limitations for various goods. Therefore, follow-up research industry-specific discussions can be made.

(ii)    In this study, we mentioned the differences between micro-enterprises and small and medium-sized enterprises. Due to sample limitations, we only focused on the lack of marketing, known market positioning, and lack of international e-commerce knowledge. If there is a way to collect more samples in the future, more research and analysis can be done on the limited product type and the start-up type.

(iii)   Due to the project scenario, the interview's content had to be separated into two portions for analysis. The integration of reality is the first section's focus, however reality plays a minimal role in this study, while the second section relates to the primary

research axis. Therefore, the interviewed companies may have some preconceived ideas, so they cannot accurately grasp the essence of the questions and answer them, resulting in some viewpoints being hidden.

### 5.4. Future Research Proposals

This study mainly focuses on small, medium, and micro-enterprise factors that can influence a firm's growth on cross-border e-commerce. In addition, future research should be used as follows:

(i) This study is a case study method. A more complete questionnaire structure can be designed and used to verify a quantitative way and complement the subjective discussion of this case study.

(ii) For the cross-border start-up type of this study, the characteristics of low product capability, low marketing capability, and low knowledge capability suggest looking for more samples to analyze and discuss providing micro-enterprises on cross-border e-commerce.

(iii) After sorting out the literature, this study will consider the factors to classify economic, social, technological, and legal factors, and suggests that more complete and more detailed considerations may be included in the scope of the study.

**Author Contributions:** Conceptualization, W.-H.C. and A.B.; Visualization, C.-L.C.; investigation, W.-H.C., Y.-C.L., A.B. and C.-L.C.; formal analysis, W.-H.C., writing–original draft, W.-H.C., validation, C.-L.C.; writing–review and editing, W.-H.C., Y.-C.L., A.B. and C.-L.C. All authors have read and agreed to the published version of the manuscript.

**Funding:** This research received no external funding.

**Institutional Review Board Statement:** Not applicable.

**Informed Consent Statement:** Not applicable.

**Data Availability Statement:** Not applicable.

**Acknowledgments:** The authors thank all case organizations and interviewees and the three anonymous reviewers for their valuable contributions to the article. The authors gratefully acknowledged the Small and Medium Enterprise Administration, Ministry of Economic Affairs of Taiwan for this research.

**Conflicts of Interest:** The authors declare no conflict of interest.

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
