# Peer review of "Influence Factors of Small and Medium-Sized Enterprises and Micro-Enterprises in the Cross-Border E-Commerce Platforms"

_jtaer, doi:10.3390/jtaer18010022_

Round 1

Reviewer 1 Report

The article contributes to the field but needs substantial improvement.

1. Motivation of the study must be included in the introduction section

2. Aspects of the sustainability need to be covered in detail. Implications are something that will after effects of the study.

3. I have not found the theoretical background in elaborated manner. Key aspects of SME and sustainability needs to be more emphasized here are a few suggestions. “An Analysis of Circular Economy Deployment in Developing Nations’ Manufacturing Sector: A Systematic State-of-the-Art Review” “Integrating Green Lean Six Sigma and Industry 4.0: A Conceptual Framework ”, “Green Lean Six Sigma critical barriers: exploration and investigation for improved sustainable performance”. This will enhance the aspects of the sustainability.

4.  There is a few typo error in the manuscript. Please fix the same.

5. Improve the referencing of the paper. There are some missing references.

6. Compare the results of your study with previous studies of the same nature.

7. Useful inferences and limitations of the study can be extended in detail.

Reviewer 2 Report

1. English Proofreading is suggested.

2. Abstract must include Covid-19, SME and sampling technique.

3. Any literary reference for 3.2 Sample and purpose?

4. More references for disasters affecting SMEs should be added.

5.  Any quotations from respondents from SME A to E?

6. Check format of Figure 3.

Good Luck !!

Reviewer 3 Report

Thank you for the opportunity to review the manuscript submitted. I enjoyed reading the manuscript and found it to be an informative study to find the influence factors of small and medium-sized enterprises and micro-enterprises in the cross-border e-commerce platforms.

Overall, the research addresses the relevant topic and takes as a starting point some questions that require further investigation. The results are appropriately interpreted and discussed.

The paper must be improved in line with the comments below:

1. The conclusion section it should also contain comparisons with various previous studies.

2. References should be updated. Please add at least 5 studies from the last 3 years.

I hope you find the above comments useful, and I wish you the best of luck with developing the paper further.

Round 2

Reviewer 1 Report

Authors have addressed all comments. The article is ready for publication.